# Deciphering the Role of Protein Phosphatases in Apicomplexa: The Future of Innovative Therapeutics?

**DOI:** 10.3390/microorganisms10030585

**Published:** 2022-03-08

**Authors:** Aline Fréville, Bénédicte Gnangnon, Asma S. Khelifa, Mathieu Gissot, Jamal Khalife, Christine Pierrot

**Affiliations:** 1Univ. Lille, CNRS, Inserm, CHU Lille, Institut Pasteur de Lille, U1019-UMR 9017-CIIL-Centre d’Infection et d’Immunité de Lille, 59000 Lille, France; bgnangnon@hsph.harvard.edu (B.G.); asmaskhelifa@gmail.com (A.S.K.); mathieu.gissot@pasteur-lille.fr (M.G.); jamal.khalife@pasteur-lille.fr (J.K.); 2Department of Infection Biology, Faculty of Infectious and Tropical Diseases, London School of Tropical Medicine and Hygiene, Keppel Street, London WC1E 7HT, UK; 3Department of Epidemiology, Center for Communicable Diseases Dynamics, Harvard TH Chan School of Public Health, Boston, MA 02115, USA

**Keywords:** Apicomplexa, *Plasmodium*, *Toxoplasma*, protein phosphatases, phosphatome, serine/threonine phosphatases, tyrosine phosphatases, PP1, anti-malaria drug design

## Abstract

Parasites belonging to the Apicomplexa phylum still represent a major public health and world-wide socioeconomic burden that is greatly amplified by the spread of resistances against known therapeutic drugs. Therefore, it is essential to provide the scientific and medical communities with innovative strategies specifically targeting these organisms. In this review, we present an overview of the diversity of the phosphatome as well as the variety of functions that phosphatases display throughout the Apicomplexan parasites’ life cycles. We also discuss how this diversity could be used for the design of innovative and specific new drugs/therapeutic strategies.

## 1. The Impact of Apicomplexa in Human Health: An Overview

Apicomplexa correspond to a large and diverse phylum of more than 6000 eukaryotic protozoa that live as obligate parasites in humans and animals [1]. Several major human pathogens such as *Plasmodium* spp. (causing Malaria), *Toxoplasma gondii* (causing Toxoplasmosis), *Cryptosporidium* spp. (causing Cryptosporidiosis) and *Babesia* spp. (causing Babesiosis) belong to this phylum. Apicomplexa are characterized by the presence of a complex of secretory organelles essential to the establishment of host cell infection, namely rhoptries and micronemes [2], and by the presence of a vestigial chloroplast-like organelle called the apicoplast, which is involved in metabolic processes crucial for parasite survival. *Cryptosporidium* is an exception, being the only Apicomplexa lacking an apicoplast [3,4,5].

*Plasmodium*, the causative agent of human malaria, is transmitted by *Anopheles* mosquitoes. It affects about half of the world population and has led to an estimated 229 million cases worldwide, mainly in Africa (93%) but also in Southeast Asia (3.4%) and the Eastern Mediterranean region (2.1%), causing major social, economic and health problems [6]. The clinical manifestations of the disease are linked to the parasite cyclic development in red blood cells and the amplitude of the host immune response. They include fever and flu-like symptoms and can lead to fatal complications such as chronic anemia or cerebral malaria. In 2019, an estimated 409,000 people died from malaria. Despite recent progress linked to local elimination campaigns launched in endemic areas and promises of the RTS,S vaccine recently recommended by the WHO, the capacity of *Plasmodium* to escape protective host immune responses and the continual emergence of resistance to current treatments and insecticides significantly impede eradication [6].

*T. gondii* is a widespread opportunistic pathogen affecting between 30% and 60% of the world population. It can infect any nucleated cell in virtually any warm-blooded animal, and its definitive hosts belong to the Felidae family [7,8]. The fetus of newly infected women during pregnancy may be subjected to severe birth defects (e.g., encephalitis and ocular diseases). *T. gondii* is also an opportunistic pathogen and reactivation of the latent forms in immunocompromised patients can lead to deadly infections [8,9]. Additionally, a growing body of evidence has recently emerged about the possible link between chronic toxoplasmosis and neurological and psychiatric conditions such as schizophrenia or bipolar disorder [10,11,12,13].

*Babesia* spp. is an intra-erythrocytic parasite transmitted by ticks. *Babesia microti* and *Babesia divergens* are the two species predominantly affecting humans, causing flu-like illness and hemolytic anemia. The elderly, immunocompromised or asplenic patients are the most at risk of developing this symptomatic disease [14].

*Cryptosporidium* is a common pathogen causing moderate-to-severe diarrhea in humans. Two species, *Cryptosporidium parvum* and *Cryptosporidium hominis*, are responsible for most cryptosporidiosis cases [15]. The disease can lead to developmental delays and malnourishment, and even death, in children aged 6 to 18 months [16,17]. Cryptosporidiosis is the leading protozoan cause of diarrheal mortality worldwide. Patients with immunocompromised systems are also at risk of developing chronic infection leading to debilitating diarrhea, which may eventually be fatal [18]. Further, cryptosporidiosis may have a causal link with digestive cancers. However, this remains under scrutiny [19].

Apicomplexa are characterized by their diversity, which is tightly linked to the way they evolved to infect specific hosts [20]. Their life cycles are complex, as they require transitions through multiple life stages. These processes can only be achieved through their ability to adjust to changing environments, escape the host defense mechanisms, and undergo massive morphological and metabolic changes [21]. A growing number of studies have recently highlighted the essential role played by phosphatases in the regulation of these processes.

In this review, we discuss these recent advances and how Apicomplexan phosphatases could prove essential to refueling research strategies and providing innovative ways of inhibiting the growth and/or the virulence of these life-threatening parasites.

## 2. Apicomplexan Serine/Threonine and Tyrosine Phosphatome

Protein phosphorylation is an ancient-in-origin post-translational modification that is probably universal across phyla. Protein kinases and phosphatases cover between 2% and 4% of a typical eukaryote’s proteome (reviewed in [22]). Unlike protein kinases that catalyze the formation of a covalent bond between a protein substrate and a phosphate group, protein phosphatases catalyze the removal of that phosphate group by hydrolysis. Another key difference between these two types of enzymes is linked to their structure; while kinases fold according to a single structure model [23], the folding of phosphatases differs according to their catalytic activities. Hence, phosphatases can be grouped into superfamilies [24]. Here, we compare the serine/threonine (S/T) and tyrosine (Y) phosphatome of several Apicomplexa of interest (namely, *Plasmodium falciparum*, *Babesia divergens*, *Toxoplasma gondii* and *Cryptosporidium parvum*) to the S/T/Y human phosphatome. We chose to focus on the enzymes capable of dephosphorylating the S/T/Y residues of proteins, as they represent most of the phosphatome of interest and are well characterized. By comparing the host and parasite phosphatome, we aim to highlight the divergences in these enzymes, which are promising sources of potential drug targets.

### 2.1. Key Differences between the Apicomplexan and Human Phosphatome

When compared to the human S/T/Y phosphatome (140 enzymes), the size of the Apicomplexan S/T/Y phosphatome is much smaller. *T. gondii* has the biggest phosphatome among the Apicomplexa considered (77 enzymes) and is followed by *Plasmodium falciparum* (40 enzymes), *Cryptosporidium parvum* (35 enzymes) and finally *Babesia divergens* (20 enzymes, see Figure 1). This is in accordance with studies led on the kinomes of these parasites, which found that Apicomplexa parasites presented the smallest kinomes among eukaryotes (from 35 sequences in *Babesia bovis* up to 135 sequences in *Toxoplasma gondii*, reviewed in [25]). The reduced number of phosphatases in Apicomplexa as compared to mammalian species is thought to result from the adaptation to a parasitic lifestyle, as parasites can live on nutrients provided by their hosts and thus require less complex metabolic regulation networks (discussed in [26]).

The composition of the protein phosphatome also differs dramatically between the parasites and their intermediate host. The most striking difference is the fact that serine/threonine phosphatases constitute around 80% of the Apicomplexan protein phosphatome but represent only around 30% of the human phosphatome, which contains mostly tyrosine phosphatases (Figure 1). It is also striking that some families of phosphatases are well conserved, while others are parasite specific. These discrepancies are detailed below.

### 2.2. General Characteristics of the Apicomplexan Serine/Threonine Phosphatome

According to the classification published by Chen et al. [24], serine/threonine phosphatases can be categorized into three major folds or superfamilies in Apicomplexa, PPPL (phosphoprotein phosphatases-like), PPM/PP2C (protein phosphatase Mn^2+^ or Mg^2+^ dependent) and HAD (haloacid dehydrogenase) (see Table 1, Figure 1, and Appendix A).

#### 2.2.1. PPPL Superfamily

Members of the PPPL fold can be sub-divided between the PPP (phosphoprotein phosphatases) and PAP (purple acid phosphatase) families.

In Apicomplexa, the PPPL superfamily is relatively well conserved across the phyla. The two members of the PAP family have been well characterized in *Plasmodium* (see later in this review). PPP members are relatively well characterized as well since they are extremely well-conserved across eukaryotes. These phosphatases tend to be expressed as catalytic subunits capable of interacting with many protein partners regulating their localization and/or activity. PPP phosphatases can be grouped into seven subfamilies: PP1, PP2A, PP2B, PP4, PP5, PP6 and PP7. *T. gondii* and *P. falciparum* possess at least a member in each of these subfamilies. Conversely, *C. parvum* does not possess any PP6 and PP7 members, and *B. divergens* lacks PP2B and PP6 (see Appendix A [27]).

Two additional classes that are absent from the human host can be observed in Apicomplexa: PPKL (Kelch-like domain-containing protein phosphatase) and SHLP (Shewanella-like protein phosphatase). *P. falciparum*, *T. gondii* and *C. parvum* possess members belonging to PPKL and SHLP classes. However, *B. divergens* only possesses one PPKL (see Appendix A).

#### 2.2.2. PPM (Protein Phosphatase Mn^2+^ or Mg^2+^-Dependent or PP2C) Superfamily

While many members of the PPP family tend to be expressed as holoenzymes, members of the PPM family rely on regulatory domains and motifs that modulate substrate specificity (reviewed in [22]) and on the binding of manganese/magnesium ions (Mn^2+^/Mg^2+^) for activity. PPM and PPPL families differ at the primary sequence level but share similarities in the way that their catalytic domains fold and catalyze the dephosphorylation of substrates (reviewed in [27]). In Apicomplexa, *P. falciparum* possesses 13 PPM members, *C. parvum* 10, and *B. divergens* 4. *T. gondii* possesses 34 members, which represents the biggest group of S/T phosphatases in this organism and exceeds in number its human host (20 PPM). Strikingly, an important proportion of the PPM family members do not have any human ortholog (i.e., 77% in *P. falciparum,* 88% in *T. gondii*, 80% in *C. parvum* and 50% in *B. divergens*, Appendix A, [28]), targeting them as potential drug targets.

#### 2.2.3. HAD Superfamily

Members of the HAD superfamily can be sub-categorized into FCP (TFIIF-associating C-terminal domain (CTD) phosphatase) and NIF (NLI interacting factor-like phosphatase) families (more families exist in non-Apicomplexan eukaryotes, see [29] for a detailed classification). Studies led on human and yeast members of these families have shown that they are involved in the regulation of highly specific dephosphorylation processes, as they target a unique substrate, RNA Polymerase II, and are also involved in interactions with transcription factor TFIIF (reviewed in [28]).

### 2.3. General Characteristics of the Apicomplexan Tyrosine Phosphatome

While the proportion of tyrosine phosphatases in the human phosphatome is greater than 65%, it is only about 10–16% in Apicomplexa. This is a rather striking observation. Although phosphorylated tyrosines have been detected in *Plasmodium* and *Toxoplasma* by mass spectrometry [30], no tyrosine kinases have been found in Apicomplexa [25]. It is thought that tyrosine kinases have emerged in unicellular eukaryotes from serine/threonine kinases, and that tyrosine phosphorylation was a means for these organisms to better sense their environment (reviewed in [31]). It is likely that tyrosine phosphatases have emerged alongside their kinase counterparts and have allowed the eukaryotic to set up new cell–cell signaling pathways [24]. It is therefore not surprising that deregulations of the expression or catalytic activity of these phosphatases can lead to disruption of immunity [32,33], organ dysfunction such as heart failure [34], or diseases such as cancer [35,36,37]. It is thus astonishing to observe that Apicomplexa have found ways to minimize the use of tyrosine phosphorylation. 

Tyrosine phosphatases are categorized into four superfamilies/folds: CC1 (cysteine-based Class I), CC2 (cysteine-based Class II), CC3 (cysteine-based Class III) and some enzymes of the HAD folds (EYA family–EYes Absent; see Table 1, Figure 1 and Appendix A). Contrary to the four human EYA phosphatases whose catalytic activity is aspartate-based, the members of the CC1-3 folds are defined by a conserved CX5R motif located in a phosphate-binding pocket (reviewed in [38]).

#### 2.3.1. CC1 Superfamily

CC1 represents the biggest group of phosphatases in humans (Figure 1) and can be sub-divided into PTP (protein tyrosine phosphatase) and the DSP (dual-specificity phosphatase) families. *P. falciparum* has two PTPs, one DSP and two RHOD DSPs (DSPs with a Rhodanese domain). *T. gondii* has one PTP, seven DSPs and four RHOD DSPs. *C. parvum* does not have any PTP but has five DSPs. Finally, *B. divergens* has one PTP and one DSP.

In Apicomplexa, little is known about tyrosine phosphatases. To date, only two of them have been enzymatically characterized. *Pf*PRL (protein of regenerating liver), a member of the *Plasmodium* PTPs, was shown in vitro to be inhibited by a tyrosine phosphatase inhibitor [39], and a similar in vitro study conducted on *Pf*YVH1 demonstrated its dual-specificity nature (i.e., its ability to dephosphorylate both S/T and Y residues) [40]. Finally, the atypical DSP *Pf*MKP1 was found to regulate the S/M transition during the life cycle of *P. falciparum* ([41], see later in the review). It therefore appears critical to explore, in more detail, the function of Apicomplexan tyrosine phosphatases to decipher their biological role and to identify differences with their host counterpart that could be exploited for drug design.

#### 2.3.2. CC2 and CC3 Superfamilies

The CC2 superfamily comprises LMWPTP (low molecular weight protein tyrosine phosphatase) and SSU72 (RNA polymerase II subunit A C-terminal domain phosphatase) phosphatases in humans. One or two LMWPTP can be found in the phosphatome of *P. falciparum* (*Pf*3D7_1127000), *T. gondii* (TGME49_305790, TGME49_232620) and *C. parvum* (cgd6_3570, cgd8_1490), and there is no CC2 in *B. divergens*’ phosphatome. CC3, which includes one family of phosphatases in humans, CDC25 (M-phase inducer phosphatase), is absent from the Apicomplexan phosphatome (Figure 1, Table 1).

#### 2.3.3. Other Tyrosine Phosphatases

Besides tyrosine phosphatases, all Apicomplexa possess one PTPL (protein tyrosine phosphatase-like), named PTPLA, which is completely absent from the host’s phosphatome (Figure 1, Table 1). This protein, localized at the endoplasmic reticulum and proved essential for mitotic division at the oocyst stage, has recently been re-annotated as DEH (3-hydroxyacyl-CoA dehydratase) and may no longer be considered as a phosphatase [42]. This highlights the need for a better characterization of all the phosphatases of Apicomplexa.

## 3. The Function of Protein Phosphatases in Apicomplexa: Where Do We Stand?

In recent years, the massive improvement of reverse genetics and multi-omics technologies has allowed for the development of ambitious projects based on global functional approaches, targeting Apicomplexa of medical importance, such as *Plasmodium falciparum* and *Toxoplasma gondii*, as well as the model organism *Plasmodium berghei*. 

Using genome-wide saturation mutagenesis (*P. falciparum*, [43]), knock-out strategies (*P. berghei*, [44]) or CRISPR screens (*T. gondii*, [45]), the contribution of each gene during *Plasmodium* asexual development or *T. gondii* development in human fibroblasts was assessed. More than 50% of *P. falciparum* and 30% of *P. berghei* phosphatases seem essential for parasite survival in blood cells, highlighting how critical these enzymes are (Figure 2, Appendix A). These studies, along with the characterization of engineered parasite mutant strains, drastically improved our understanding of the critical role played by protein phosphatases over the life cycle of these parasites.

In the following section of this review, we summarized our current knowledge regarding the functions of the S/T/Y phosphatases in *Plasmodium* and *Toxoplasma*.

### 3.1. Functional Characterization of Plasmodium Proteins Phosphatases

#### 3.1.1. PPPL Fold

Phosphoprotein Phosphatases (PPP) family

##### Protein Phosphatase Type 1 (PP1)

Protein phosphatase type 1 is an extremely well-conserved phosphatase known to be involved in a myriad of biological processes such as protein synthesis, transcription regulation, and cell division [46]. The PP1 catalytic domain possesses the distinct characteristic of interaction capacity with hundreds of regulatory subunits, thus leading to the formation of numerous holoenzymes, and allowing the enzyme to be involved in the regulation of a wide range of biological processes [47]. In *Plasmodium falciparum*, *Pf*PP1 is expressed throughout the intraerythrocytic development cycle (IDC), mainly from the trophozoite to the schizont stage [48,49,50]. *Pf*PP1 has been localized at the parasite’s nucleo-cytoplasmic compartment [48], but also at the Maurer’s clefts where it was suggested to modulate the phosphorylation of *Pf*SBP1, an erythrocyte skeletal-binding protein [51]. The first evidence of the enzyme’s massive impact on the phosphorylation levels in the parasite came from early studies showing the drastic effect of phosphatase inhibitors on parasite development [52]. These observations were later confirmed by a reverse genetics screen, suggesting its essentiality for IDC completion [43]. Recently, inducible knockdown (iKD) approaches were used to refine *Pf*PP1 function. These studies revealed a role in DNA replication, the formation of multinucleate schizonts, and egress from the host red blood cell. This last observation was linked to the parasite’s inability to secrete *Pf*SUB1, a protease essential to the orchestration of the early steps of egress when *Pf*PP1 is absent [50] (Figure 3, Appendix A).

To assess the function of PP1 during development in mosquito and liver cells, various reverse genetics strategies were implemented using the *Plasmodium berghei* model of rodent infection. When a reduction of *pbpp1* expression in gametocytes was engineered following a promotor swap approach, drastic effects on male exflagellation, ookinete conversion, and a complete inability to transmit the infection were observed [53]. When pbpp1 was conditionally disrupted from the sporozoite stage (FlpL/FRT site-specific recombination system, [54]), no effect was observed on sporozoite formation or development in liver cells. However, merozoites exiting hepatocytes lost their ability to spread the infection, thus confirming PP1 essentiality for parasite development in the blood ([43], Figure 4, Appendix A).

Comparative sequence analysis, in silico motifs, and yeast two-hybrid screening of cDNA libraries shed light on the existence of hundreds of potential *Plasmodium* PP1 regulators [55]. Among them, four conserved regulators previously described in mammals, as well as three *Pf*PP1 regulators restricted to the *Plasmodium* genus, have been characterized for their ability to bind and regulate the phosphatase activity [48,49,56,57,58,59,60,61,62] (for more details on PP1 and its interactors in *Plasmodium*, see also review [63]).

##### Calcineurin (PP2B; CnA, CnB)

Calcineurin has been previously described to be involved in numerous processes related to calcium signal transduction [64]. PP2B exists as a holoenzyme formed by the association of CnA, a catalytic subunit with CnB, its regulatory subunit [65]. In *P. falciparum*, the phosphatase was first purified from the cytosolic extract and identified due to its susceptibility to cyclosporin A (CsA) and FK506 [66], two calcineurin inhibitors described in mammalian cells [67]. Later, these two compounds were shown to inhibit the secretion of microneme proteins, suggesting a link between the phosphatase and calcium-dependent microneme exocytosis [68]. However, when using a conditional protein expression system targeting CnB, in a study carried out by Paul AS et al., it was demonstrated that, although calcineurin is essential to the parasite development in erythrocytes, its function is independent of microneme exocytosis but is rather related to a poorly understood form of strong attachment to the red blood cell before invasion [69]. Accordingly, by using an auxin-inducible degron (AID) system in *P. berghei*, calcineurin’s (CnA) critical function on merozoite attachment and invasion was confirmed. Further analysis during mosquito development revealed essential functions for gamete development and fertilization, but also in ookinete infectivity and sporozoite invasion ([70], Figure 3 and Figure 4, Appendix A).

##### PP5

Unlike other phosphatases belonging to the PPP family, PP5 is a monomeric phosphatase containing both catalytic and regulatory subunits. This class of enzyme is also characterized by the presence of N-terminal tetratricopeptide repeat (TRP) domains known to be involved in autoregulation and in protein–protein interactions [71,72,73]. In *P. falciparum*, PP5 possesses structural and functional similarities with its orthologs. Its N-terminal TPR region seems to be required for the phosphatase’s auto-inhibition, and similar to mammalian cells, can interact with heat shock protein 90 (hsp90) [73,74,75]. A series of reverse genetic analyses leading to the GFP tagging of *P. berghei* PP5 and its gene deletion revealed that the enzyme is highly expressed in male gametocytes and is critical for the regulation of male fertility ([76,77], Figure 4, Appendix A).

##### Protein Phosphatase Containing Kelch-like Domains (PPKL)

The protein phosphatase containing Kelch-like domains (or PPKL) is intriguing as it belongs to a family that can only be found in plants and alveolates [78]. Therefore, PPKL appears as an interesting drug target. The phosphatase was initially identified in *P. falciparum* (under the name PPα) and, early on, was suggested to be involved in sexual stages development, as its transcript could only be detected in gametocytes [79]. This observation was later confirmed using the *P. berghei* model. Knock-out reverse genetics analyses revealed that *pbppkl* deletion severally impacts ookinete morphology and the ability to glide, as it resulted in parasites that could not develop in the mosquito midgut and, therefore, could not be transmitted ([80,81], Figure 4, Appendix A).

##### *Schewanella*-like Phosphatases

“*Schewanella*-like phosphatases” (or Shelphs/SHLPs) belong to a class of enzymes closely related to phosphatases found in the bacteria genus Schewanella [82]. Besides in bacteria, Shelphs proteins can be found in fungi, plantae, and Chromoalveolata and Excavata parasites [83]. Therefore, they represent an interesting target for drug design. In *Plasmodium*, two genes encoding *Schewanella*-like phosphatases have been identified: *shlp1* and *shlp2*. In *P. falciparum*, *shlp1* was described as likely essential for erythrocyte development by a functional screen analysis [43]. On the contrary, in *P. berghei*, the gene is dispensable at that stage, but its knock-out impacts the parasite’s ability to produce proper ookinetes, ultimately resulting in a drastic loss of the oocyst formation [83]. Conversely, *shlp2* knock-out does not seem to disturb the parasite erythrocyte development in either species [44,76,84] (Figure 3 and Figure 4, Appendix A), although the early observation of its apical localization in *P. falciparum* merozoites first suggested a role in invasion [85].

2.Purple acid phosphatase (PAP) family

##### GAP50/SAP (Secreted Acid Phosphatase)

In *P. falciparum*, GAP50 (or glideosome-associated protein 50) was first identified as part of the MyoA–MTIP–GAP motor complex or glideosome: an Apicomplexan complex involved in gliding and invasion [86]. In *Plasmodium* merozoites, the protein was, as expected, localized to the IMC (inner membrane complex), where it participates as a membrane anchor linking the glideosome to the IMC and was suggested to take part in erythrocyte invasion [87,88]. Subsequently, the protein was also detected on the surface of gametocytes [89,90]. In the mosquito midgut, GAP50 is involved in the co-opt of Human Factor H, allowing the parasite to evade complement-mediated lysis [90].

The resolution of the *Pf*GAP50 crystal structure revealed similarity to purple acid phosphatases (PAPs) [91]. This confirmed an earlier study describing the protein as an acidic phosphatase (renamed SAP) with a broad substrate profile, secreted to the parasite periphery where it was proposed to be involved in host nutrient acquisition through dephosphorylation [92].

##### UIS2

In *Plasmodium* salivary gland sporozoites, protein synthesis is inhibited by the phosphorylation of a translational factor: Eif2α. Any further development into the host liver requires its dephosphorylation. In mammalian cells, Eif2α is regulated by PP1 [93], but it does not seem to be the case in *Plasmodium*. Instead, the parasite relies on UIS2, a phosphatase restricted to Apicomplexa, to ensure the proper regulation of Eif2α and the timely translation of genes essential to the sporozoite transformation into exo-erythrocytic stages. UIS2 activity can be enhanced by Mn^2+^ and inhibited by Cd^2+^ but remains unchanged by the okadaic acid (PP1/PP2A phosphatase inhibitor). Thus, the phosphatase was first classified as a PP2C/PPM family member ([94], (Figure 4, Appendix A). However, the protein does not possess a canonical PP2C catalytic domain but shares some similarity with purple acid phosphatases and, therefore, was classified as a PAP in this review.

#### 3.1.2. Protein Phosphatase Mn^2+^ or Mg^2+^ Dependent (PPM) Fold

3.Protein Phosphatases Mn^2+^ or Mg^2+^ dependent (PPM) family

Most of the information related to the function of *Plasmodium* PPM came from genome-wide functional studies carried out using the *Plasmodium berghei* model. In an earlier analysis aiming to target the parasite protein phosphatome, the knock-out of 9 of the 13 existing *Pb*PPMs (Table 1, Appendix A) was attempted [76]. According to this study, two genes (*ppm6* and *ppm9)* were described as essential for the completion of parasite development in blood, and four genes (*ppm3, ppm6, ppm7, and ppm8)* were described as dispensable for development in both the murine host and mosquito vector. Finally, three genes (*ppm1*, *ppm2* and *ppm5*) were analyzed for their implication in the development in mosquitos. *Ppm1* knock-out mutants formed defective, non-exflagellating male gametocytes. *Ppm2* mutants, conversely, produced fewer female gametocytes. The ookinete conversion, blocked at early retort stage II, was also drastically impacted. Finally, *Ppm5* knock-out parasites produced few oocysts that were unable to form sporozoites, suggesting a role in the regulation of the oocyst development. Ookinete ultrastructure analysis revealed a partial defect in microneme formation, which could be responsible for the observed phenotype [76].

More recently, a genome-wide functional screen analysis nuanced the phenotype data described above. According to this study, *Pb*PPM1 and *Pb*PPM2 may have a mild role in parasite growth, and *Pb*PPM9 may be dispensable at this stage. Additionally, the remaining members of the PPM family: *Pb*PPM10, *Pb*PPM12 and *Pb*PPMX1, were suggested to be dispensable for the development in erythrocytes ([44], see the comparison of both analysis outcomes in Appendix A). In *Plasmodium falciparum*, *Pf*PPM2 contains two catalytic domains whose dimerization is required for optimal activity [95]. Further analyses were performed to identify the phosphatase substrates. The translation elongation factor 1-beta was identified among other components of the translation and transcription machinery [96], suggesting that *Pf*PPM2 may play a role in the regulation of those mechanisms. Using a saturation mutagenesis approach, Zhang et al. carried out a functional screen revealing that *Pf*PPM2, *Pf*PPM6, *Pf*PPM8 and *Pf*PPM9 could be essential for the parasite blood development stage when *Pf*PPM3 and *Pf*PPM10 play a mild role [43]. On the contrary, the remaining members of the family (*Pb*PPM1, *Pb*PPM4, *Pb*PPM5, *Pb*PPM7, *Pb*PPM11, *Pb*PPM12, and *Pb*PPMX1) were described as dispensable (Figure 3 and Figure 4, Appendix A). 

The comparison of the functional screening performed in both *falciparum* and *berghei* models revealed some discrepancies suggesting that species specificity may be at play here.

4.*Pf*MKP1

*Pf*MKP1 was identified and classified based on some structural similarities with mitogen-activated protein kinase (MAPK) phosphatases (or MKPs) which, in other organisms, have been described in cell cycle progression, cell growth and proliferation [97]. *Pf*MKP1 sequence analysis revealed the presence of a non-catalytic N-terminal rhodanese (RDH)-like domain upstream of a catalytic dual-specificity phosphatases (DUSP)-like region, an association characteristic of MKPs [98,99,100]. However, the poor conservation of some critical catalytic residues suggests that the protein may be of low activity or may be a pseudo-phosphatase. Using random insertional mutagenesis, Balu et al. observed that the presence of a single transposon downstream of the *mkp1* start codon of a *P. falciparum* cloned line was responsible for a significant attenuation of the parasite growth. Subsequent analysis revealed that the phenotype was due to a prolonged pre-S phase (trophozoite stage) causing a delayed entry into the S/M phase (schizont stage) ([41], Figure 3, Appendix A).

### 3.2. Functional Characterization of Toxoplasma gondii Proteins Phosphatases

#### 3.2.1. PPPL Fold

5.Phosphoprotein Phosphatases (PPP) family

Few protein phosphatases belonging to the phosphoprotein phosphatase family (PPP) have been characterized in *Toxoplasma gondii* thus far. However, the presence of members belonging to the PPP family has been established. Relatively comparing the genome of eukaryotes to that of *T. gondii* has demonstrated that the latter possesses a minimum of one member belonging to each of the seven PPP sub-families. Thus, the *T. gondii* genome encodes a complete set of PPP sub-families including PP1, PP2A, PP2B (calcineurin), PP4, PP5, PP6, and PP7 [27].

##### Protein Phosphatase Type 1 (PP1)

*T. gondii* PP1 (*Tg*PP1) has been studied to a lesser extent compared to PP1 from other eukaryotes and the Apicomplexan parasite *Plasmodium falciparum*. *Tg*PP1 was initially identified through experiments in which *T. gondii* tachyzoites were exposed to specific inhibitors of PP1, including tautomycin (TAU) and okadaic acid (OA). Tachyzoites treated with TAU and OA demonstrated a defect in invasion by a percentage of 50%. The impairment of invasiveness of *T. gondii* tachyzoites suggested that *Tg*PP1 has an important role in the invasion of the host cell [101]. However, these inhibitors could target other phosphatases, and the effect demonstrated may be due to the inhibition of multiple phosphatases. Studies carried out on *Tg*PP1 involving phosphatase assays on extracts of *T. gondii* have indicated the presence of *Tg*PP1 dephosphorylation function, which is sensitive to specific *Tg*PP1 inhibitors such as human PP1 Inhibitor 2 and okadaic acid [101]. Attempts to identify specific protein targets of *Tg*PP1 were carried out by means of immunoprecipitation assays as well as in vitro phosphate labeling assays [101] but yielded few potential targets. Among *Tg*PP1 interactors, *Tg*LRR1, a homolog of sds22 in yeast, was identified to form a complex with *Tg*PP1. The association of *Tg*PP1 with *Tg*LRR1 was demonstrated to occur within the nucleus of the tachyzoite [102]. *Tg*LRR1 was identified as an inhibitor of *Tg*PP1 by using the *Xenopus* oocyte model during which germinal vesicle breakdown takes place once the okadaic acid inhibitor or *Tg*PP1 antibodies are micro-injected within the oocytes [102]. Furthermore, the micro-injection of oocytes with cRNA of *Tg*LRR1 led to nearly a total germinal vesicle breakdown of oocytes, suggesting that *Tg*LRR1 can inhibit the *Xenopus* PP1. A more recent study regarding inhibitors of *Tg*PP1characterized inhibitor-2 (*Tg*I2) that it functions similarly to its eukaryotic homologs exhibiting inhibitory activity [103]. The *Tg*I2 inhibitor has three different motifs (SILK-like, RVxF, and FKK/HYNE). *Tg*PP1 associates with *Tg*I2 through the RVxF motif, whereas other motifs play a more minor role in the binding to *Tg*PP1 [103]. However, both SILK-like and RVxF motifs are critical for regulating the activity of *Tg*PP1, a feature that is common with the higher eukaryote’s Inhibitor-2 protein but was not observed for *Pf*PP1 [103]. Relatively little has been described regarding *Tg*PP1 biological function. Although it has been suggested to be essential during infection of human cells ([45], Appendix A), many questions remain unanswered, such as its specific mechanism of action and its specific targets.

##### PP2A and PP2B (Calcineurin)

Another protein belonging to this family of phosphoprotein phosphatases is PP2A. PP2A has been shown to have a key role in regulating the cell cycle in eukaryotes and remains crucial for regulating the progression from the G2 phase to the M phase. The deletion of PP2A led to a delayed cell cycle and an impairment of the spindle assembly checkpoint [104]. Despite the central role of PP2A in the cell cycle of mammals and higher eukaryotes and its existence within *Toxoplasma,* not much is known about its function within the Apicomplexan parasite *T. gondii* and remains to be characterized in future studies.

Recently, conditional knock-down approaches have been implemented to decipher the role of calcineurin in *Toxoplasma gondii*. The phosphatase seems, similar to its *Plasmodium falciparum* counterpart, to be involved in the regulation of host cell attachment independently of microneme exocytosis ([69], Figure 5, Appendix A).

6.Purple acid Phosphatase (PAP) family

##### GAP50/SAP (Secreted Acid Phosphatase)

Similar to *Plasmodium falciparum*, *Tg*GAP50 is a critical element of the glideosome. Harding et al. demonstrated that gap50 conditionally depleted the parasites’ present defects in the IMC morphology and the localization of the glideosome’s components, suggesting a role in IMC biogenesis and stability during parasite intracellular replication ([105], Figure 5, Appendix A).

##### GRA44 (TGGT1_228170 Secreted Acid Phosphatase)

GRA44 was initially identified as a secreted acid phosphatase based on a bioinformatics analysis of proteins within the *T*. *gondii* genome, containing predicted phosphatase domains and signal peptides. Localization studies of GRA44 identified that this acid phosphatase localized within the PV lumen and at the PV membrane, where two specific amino acids were demonstrated as necessary for its proper processing, which is not essential for its function [106]. Additionally, the conditional knockdown of GRA44 allowed it to demonstrate significantly impaired growth in the mutant parasites compared to the parental strain. GRA44 was demonstrated to associate with components of the putative translocon complex, which includes MYR1, MYR2, and MYR3 and induces c-Myc expression, which is crucial for the translocation of the parasite’s effector proteins [106], Figure 5, Appendix A).

#### 3.2.2. Protein Phosphatase Mn^2+^ or Mg^2+^ Dependent (PPM) Fold

7.Protein Phosphatases Mn^2+^ or Mg^2+^ dependent (PPM) family

Most of the work related to the functional analysis of *Toxoplasma gondii* phosphatases has been performed on proteins belonging to the PPM family. Recently, Yang et al. carried out the analysis of the sub-cellular localization of five of them (PPM2A, PPM2B, PPM3D, PPM5 and PPM11), revealing an intriguing diversity of localizations [107]. TgPPM2A was found to be associated with the nucleocytoplasmic compartment. TgPPM2B, conversely, is expressed in the cytoplasm but is excluded from the nucleus. TgPPM3D, albeit predicted to contain a transmembrane domain, was found to be expressed in the endoplasmic reticulum. A more detailed investigation carried out on TgPPM5C revealed that the protein, expressed at the plasma membrane, is involved in host cell attachment. TgPPM5C possesses some putative myristoylation and palmitoylation sites, specifically glycine residue 2 and cysteine residue 4, which are both required for the proper localization and function of the TgPPM5C protein at the plasma membrane. Studies carried out on TgPPM5C knock-out parasites demonstrated that these mutant parasites possess a defect in parasite propagation, which is linked to an impairment of the mutant parasite to invade the host cell ([107], Appendix A). Adhesion assays indicated that the defect in TgPPM5C knock-out parasites is in the parasite’s ability to attach to the host cell. Additionally, the protein phosphatase activity, although not directly linked with TgPPM5C localization, was proven essential to its function. Moreover, TgPPM5C was shown to modulate the phosphorylation status of multiple signaling proteins such as Rab-GTPases, multiple kinases, a guanylyl cyclase, and phosphatases. Finally, TgPPM11 is secreted in the parasitophorous vacuole. This observation will require further study, as an error in the protein annotation may be accountable for the result ([107], Figure 5, Appendix A). 

More recently, TgPPM3C was identified as a phosphatase secreted into the lumen of the parasitophorous vacuole. It has been shown to interact with MYR1, an essential protein belonging to a vacuolar transposon involved in export to the host cell [108]. *Tgppm3c* knock-out led to the production of parasites exhibiting a mild growth defect in vitro but with a profound loss of virulence during in vivo acute infection. Electron microscopy revealed an absence of major differences between the wild-type and TgPPM3C deficient parasites [109]. Furthermore, a phospho-proteomics analysis was carried out to identify potential substrates of TgPPM3C. In total, 118 phospho-peptides were more abundant in the TgPPM3C knock-out strain than in the corresponding wild-type. Alternatively, merely 10 phospho-peptides were identified as significantly less abundant [109]. The phospho-peptides belonging to significantly abundant proteins were determined to be part of the rhoptry, dense granule, and microneme compartments. A significant decrease in the abundance of TgGRA16 and TgGRA28 export to the host nucleus was observed in cells infected with the TgPPM3C knock-out strain when compared to those infected with its wild-type counterpart, suggesting an impairment in the export of these two effector proteins ([109], Figure 5, Appendix A). 

This was confirmed by carrying out phospho-mimetic mutation experiments on TgGRA16, where a decreased accumulation of this protein within the host nucleus was observed, suggesting a reduction in the export of the GRA16 effector protein. The reduction of TgGRA16 and TgGRA28 export may be linked to the virulence defect observed in vivo. Furthermore, during chronic infection, these defective parasites formed less cysts in the mouse brain that could be linked to an alteration of the phosphorylation and export status of GRA16 and GRA28 [109].

TgPPM13 (also named TgPP2C) is a nucleocytoplasmic protein that, when overexpressed, impacts parasite cytokinesis, suggesting a role in the regulation of cell cycle control [110]. TgPPM13 was also shown to regulate the phosphorylation of Toxofilin [111], a secreted protein that is known to bind host actin [112]. TgPPM13 carries out its function in accordance with a kinase known by the host casein kinase II (CKII). CKII phosphorylates Toxofilin specifically by targeting Serine 53. The association between TgPPM13 and CKII results in the sequestering of actin monomers as well as the regulation of binding between Toxofilin and host actin impacting the parasite’s movement [111,113].

Finally, TgPPM20 (also referred to as PP2C-hn) displays a mild role in parasite growth. During invasion, it is secreted from the rhoptries and is directed to the host cell nucleus, suggesting a role in host modulation [114]. Recently, this secreted phosphatase-lacking homolog in its host has triggered interest and was successfully tested as a potential vaccine candidate using a murine model [115]. However, further studies are needed in order to identify the direct targets of this PP2C protein (Figure 5, Appendix A).

8.Aspartate-based phosphatase family (FCP/SCP)

Phosphatases belonging to this family vary from members of other phosphatase families due to their characteristic aspartate catalytic activity. Most of the *T. gondii* aspartate-based phosphatases consist of a conserved motif known by DxDT/V. *T. gondii* possesses eight of these aspartate-based phosphatases. According to the literature, none of these proteins in *T. gondii* have been functionally characterized. FCP1, a single protein previously characterized in humans, belongs to this family and functions in modulating the phosphorylation status of the carboxy-terminal region of the RNA polymerase II enzyme [116]. *T. gondii* possesses the FCP1 gene as other Apicomplexa. However, within the *T. gondii* parasite, the regular CTD function observed in other Apicomplexa is absent and is most likely due to its characteristic low number of heptapeptide repeat pairs [116].

In sum, in Apicomplexa, S/T/Y phosphatase roles are diverse and seem to impact every aspect of the parasite’s asexual/sexual development and interaction with its host and environment. Therefore, these enzymes must be considered as a great potential for the design of innovative and specific treatment. In the next section of this review, we will highlight and discuss opportunities supporting this idea in *Plasmodium.*

## 4. Therapeutic Potential of Protein Phosphatases as Drug Targets: The Case of *Plasmodium*

Over the last decades, most research and development companies have focused on targeting protein kinomes (comprising more than 350 kinases) [117] to treat various human diseases, especially cancer. The search for inhibitors against kinases has been tremendously developed since these enzymes are considered specific targets, thus contributing to selective treatments decreasing the risks of adverse drug effects.

With respect to protein phosphatases, their validation as drug targets first appeared less captivating and more challenging. They were considered less specific, given the conserved nature of their catalytic subunits, and more challenging to target, due to the level of uncertainty and the paucity of data regarding their molecular functions along with some major technical bottlenecks hampering the identification of their substrates. Nevertheless, it is now well known that protein phosphatases encompass diverse types of enzymes that are involved in the control of dephosphorylation processes in an extremely precise way and coordinated manner in a broad range of vital cellular functions. Hence, it is not surprising to uncover that phosphatases, when dysregulated, are found to be associated with a plethora of human diseases, including diabetes, cancer, or neurodegeneration [118]. Therefore, phosphatases could be considered attractive therapeutic drug targets. This is supported by recent reviews reporting why and how protein phosphatases should be included in the list of druggable enzymes or proposed as clinically pertinent targets [38,119,120,121].

### 4.1. Targeting Phosphatases to Treat Human Diseases: Two Decisive Cases

In recent studies, considerable and significant progress has been made to better define the functions and contribution of phosphatase signaling in normal development and pathology. This led to breakthroughs demonstrating the feasibility of targeting those enzymes with concrete impact on human diseases. The first evidence emerged in 2016 when Chen et al. showed that a small molecule named SHP099 was able to bind and selectively block the function of the protein tyrosine phosphatase SHP2, which was reported as the first oncogenic phosphatase linked to multiple cancers [122]. This molecule has been shown to be a potent inhibitor of the growth of human cancer cells in vitro and of xenograft tumors in a mouse model when administered via the oral route. Current clinical trials are still under investigation to validate SH099 as a viable drug for the treatment of human diseases. The second example came from a study in which a specific inhibitor of a regulator of PP1 (PPP1R15B) was shown to control translation and improve proteostasis as well as deficiencies linked to Huntington’s disease in a mouse model [123]. These successful strategies have certainly contributed in 2021 to the creation of the first research and development company (Anavo Therapeutics, [124]) intending to develop drug leads targeting phosphatases with the hope to accelerate the research in the field and provide additional means for the treatment of human diseases.

### 4.2. Is Targeting Phosphatases the Future of Anti-Plasmodium Therapeutics?

More than 95% of *Plasmodium* proteins are Ser/Thr phosphorylated [125], suggesting that their dephosphorylation is a major process in the regulation of protein activities. Additionally, *Plasmodium* not only expresses conserved phosphatases whose actions may be adapted and distinct from humans but also specific phosphatases that can also be targeted with minimal adverse effects.

In this perspective, along with the fact that the identification of drugs with novel modes of action becomes decisive, we will highlight and discuss opportunities for targeting *Plasmodium* protein phosphatases for the development of drugs for malaria treatment.

### 4.3. Targeting the PPP Family

#### 4.3.1. PP1 and PP2A

While there are about 38 human Ser/Thr phosphatases, almost 90% of the cellular Ser/Thr phosphatase activities are covered by PP1 and PP2A. Although these two enzymes share a conserved catalytic subunit, they have been shown to exhibit distinct cellular activities at specific times of the cell cycle, thus indicating that they dephosphorylate specific substrates [126]. This is further substantiated by the fact that they function as holoenzymes with a great number of different regulatory subunits (transport, specificity, and control of activity of the catalytic subunits [127,128]), which could pave the way for new means to selectively interfere with phosphatases activities for effective therapy.

In *Plasmodium falciparum*, the first report highlighting the importance of phosphatases and their potential to control parasite growth came from the use of two natural toxins: okadaic acid and calyculin A. These toxins are well-known potent inhibitors of Ser/Thr mammalian phosphatases PP1 and PP2A, albeit not at the same concentration. If PP2A can be inhibited by okadaic acid and calyculin A at approximatively the same concentration, the amount of okadaic acid required to inhibit PP1 is about 100 times higher than the amount required for calyculin A in humans (reviewed in [129]). In their study, Yokoyama et al. demonstrated that, in vitro, the addition of okadaic acid or calyculin A led to a drastic inhibition of blood-stage parasite growth with an IC_50_ of 9.9 and 38.5 nM, respectively [130]. This was recently confirmed by our analysis of these toxins from which we added cantharidic acid (CA), which showed an IC_50_ of about 4.3 µM (personal communication). Based on these observations, it is likely that the inhibition of in vitro parasite growth following the use of natural toxins is due to a drastic blockage of PP1 activities.

Over the last decade, there has been a particular interest and rebound in targeting the catalytic subunit of PP1 alone or as a holoenzyme. Given the low probability to develop a selective lead or drug acting against PP1 because of the high conservation of its active site within its family and across species, we and others examined strategies aiming to target PP1 regulatory sub-units to interrupt the formation of PP1 holoenzymes [56,58,61,121,131,132,133]. The feasibility of those strategies was supported by several studies carried out on viruses showing that the use of small molecules designed to bind to the non-catalytic RVxF binding site of PP1 [134] was able to prevent viral replication in vitro of viruses such as HIV1 and Ebola [135,136,137]. One small molecule showed anti-HIV1 activity in a humanized mouse model [136]. In *P. falciparum*, we observed the potent inhibitory effect on the parasite’s in vitro growth of binding peptides derived from PP1 partners mimicking interaction sites with the phosphatase [56,58,61,63,138]. From these studies, it becomes obvious that the interference with the assembly PP1 holoenzymes can constitute a novel and major pathway to be drug targeted in *Plasmodium* (and other apicomplexan parasites).

#### 4.3.2. Calcineurin (PP2B)

Indirect evidence has demonstrated that cyclosporin A (CsA) is a potent inhibitor of *Plasmodium* calcineurin (PP2B) when found associated with *Plasmodium* cyclophilin 19A and 19B [66,139]. Cyclosporin A has been shown to inhibit *P. falciparum* growth in vitro [140,141,142]. For *P. berghei,* its administration in mice (5 mg/kg) induced a reduction in blood parasitemia associated with an increase in host survival, which may be linked to an increase in suicidal erythrocyte death (eryptosis) [143]. More recent studies have shown the presence of human cyclophilin B on the surface of red blood cells, which along with additional parasitic cyclophilins, could be possible targets of cyclosporin A, contributing to the inhibition of *Plasmodium* growth [144,145]. However, the impact of cyclosporin A on *Plasmodium* calcineurin activity requires further investigation.

#### 4.3.3. PP4 and PP6 In Silico Analysis of Druggability

Recently, Ali et al. created a list of anti-malaria potential drug targets to be prioritized for further investigation. *P. falciparum* proteins previously identified as essential for parasite blood development were subjected to in silico analysis to assess their druggability potential [146]. From this screening, two Ser/Thr phosphatases: PP6 and PP4, were identified based on their sub-cellular localization, their structure, and the absence of host homologs. PP4 was even shortlisted as one of the most promising potential targets due to some promising structural features such as druggable pockets or the fact that it may be an important component of the *Plasmodium* metabolic network [146].

### 4.4. Targeting the Tyrosine Phosphatases

#### 4.4.1. In Silico Docking Analysis of PRL

In a recent study, Pandey et al. carried out an in silico virtual screening of the ChEMBL-NTD library using the *Plasmodium falciparum* phosphatase of regenerating liver (*Pf*PRL) as a drug target [147]. Among the identified potential hits, they demonstrated that the Novartis_003209 compound was able to inhibit the phosphatase activity of recombinant *Pf*PRL and was able to prevent the growth of intraerythrocytic *P. falciparum* parasites in vitro (IC_50_: 0.273 µM) [148]. This compound does not display any cytotoxicity against human cells [149] and therefore should be considered as a potential antimalarial. Further in silico docking analysis performed on other *Plasmodium* tyrosine phosphatases (*Pf*YVH1, *Pf*3D7_112700 and *Pf*PTP1) suggested that Novartis_003209 can bind to their conserved catalytic domain. Based on these data, it will be interesting to use the structural scaffold of this hit, and to pharmacomodulate it to design a compound library that will allow for the identification of specific tyrosine phosphatases inhibitors with improved potency and selectivity against *Plasmodium* in vitro and in vivo development.

#### 4.4.2. In Silico Docking Analysis of PfMKP1

A homology model of *Pf*MKP1 was created using a crystal of human MKP3 [150], with whom it shares about 21% similarity. The model was then used to screen antimalarial compounds from the ChEMBL-NTD library. The IC_50_ of seven compounds showing a high binding affinity with *Pf*MKP1 in silico was measured in vitro against WT and ∆*Pf*MKP1 *P. falciparum* parasites. For three of them, the IC_50_ was significantly higher in the ∆*Pf*MKP1 parasites, suggesting that the phosphatase may be a potential target for these compounds. As a whole, this study represents a good example of the use of structure-based drug discovery strategies as a new means to identify new classes of antimalarial drugs [151].

## 5. Conclusions and Future Perspectives

Despite recent progress, diseases caused by Apicomplexan parasites remain of human and veterinary importance. The rapid spread of emerging resistance to current anti-malaria front-line drugs has only urged for innovative therapeutics. In this review, we attempted to summarize the current knowledge regarding Apicomplexan phosphatase diversity in terms of their structure, regulation, and function. We also discussed how their most distinct features could prove essential in the design of innovative drugs. As an example, we pointed out the combinatory capacity of some phosphatases’ catalytic sub-units to create potent interfering peptides.

In recent years, tremendous progress has been made to elucidate Apicomplexan phosphatase function. These studies, mostly carried out in *Plasmodium*, highlighted the crucial role of some specific phosphatases, revealing opportunities for interventions. In-depth analysis will be necessary to explore them further. Nevertheless, several research groups have already based their recent studies on both functional analyses and subsequent in silico structural analyses to successfully screen some compound libraries for anti-malarial purposes. In this context, the recent availability of the new AlphaFold Protein Structure Database version (https://alphafold.ebi.ac.uk, accessed on 20 January 2022), allowing for highly reliable prediction of a protein structure, will undoubtedly accelerate the in silico based drug design. This highlights, once again, the importance of multidisciplinary joint efforts in the fight against diseases caused by Apicomplexa.

## Figures and Tables

**Figure 1 microorganisms-10-00585-f001:**
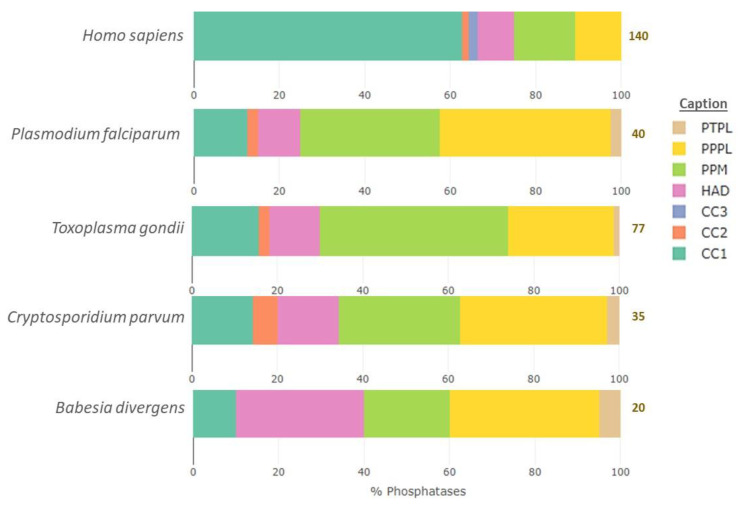
Comparison of the human S/T/Y phosphatome percent composition to those of *Plasmodium falciparum*, *Toxoplasma gondii, Cryptosporidium parvum* and *Babesia divergens*. The total number of phosphatases included for each organism is indicated on the right-hand side of each bar. PTPL, protein tyrosine phosphatase-like; PPPL, phosphoprotein phosphatase like; PPM, protein phosphatases Mn^2+^ or Mg^2+^ dependent; HAD, haloacid dehydrogenase; CC1-3, cysteine-based Class I-III. The figure was made using plotly in R version 3.6.3.

**Figure 2 microorganisms-10-00585-f002:**
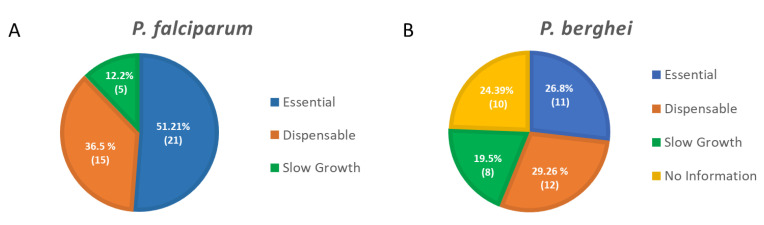
(**A**). Pie chart of the protein phosphatases’ essentiality in *P. falciparum* for the parasite blood stage development as determined in the genome-wide saturation mutagenesis [43] (See also Appendix A). (**B**). Pie chart of the protein phosphatases’ essentiality in *P. berghei* for the parasite blood stage development as determined in the PlasmoGEM study [44] (see also Appendix A). Essential, Dispensable, and Slow Growth represent relative growth rate of 0.1, 1.0, and between 0.1 and 1.0, respectively. The overall percentage and corresponding number of protein phosphatases identified as essential, dispensable or with a slow growth phenotype are represented in the pie charts.

**Figure 3 microorganisms-10-00585-f003:**
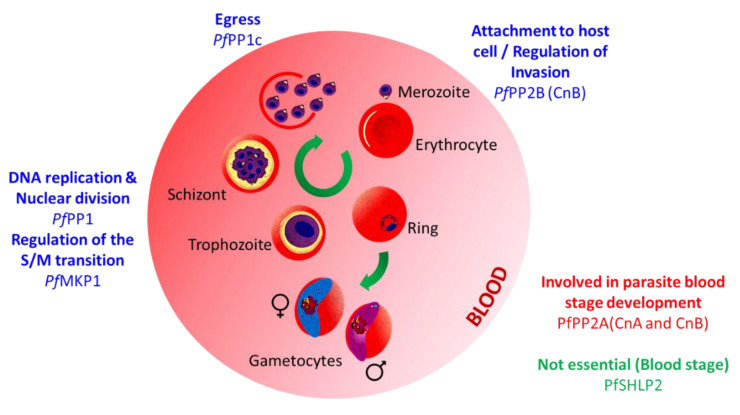
Schematic illustration of *P. falciparum* development in red blood cell showing the protein phosphatases with known stage specific function (see Appendix A, Appendix A for details and references). This scheme was created using PowerPoint from Microsoft Office 365.

**Figure 4 microorganisms-10-00585-f004:**
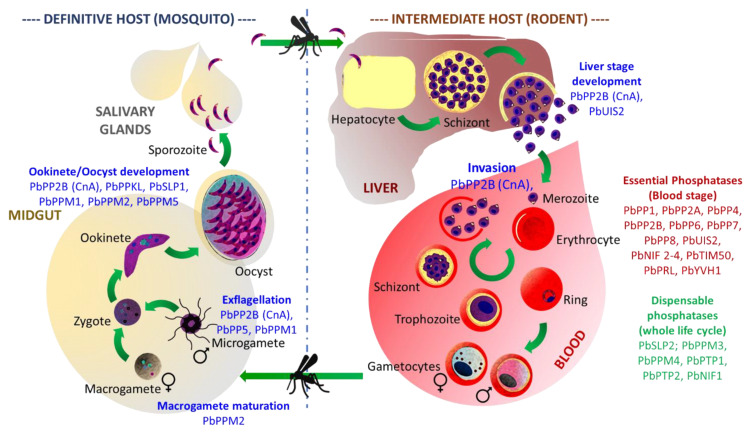
Schematic illustration of *P. berghei* life cycle showing the protein phosphatases with known stage specific function (see Appendix A, Appendix A for details and references). This scheme was created using PowerPoint from Microsoft Office 365.

**Figure 5 microorganisms-10-00585-f005:**
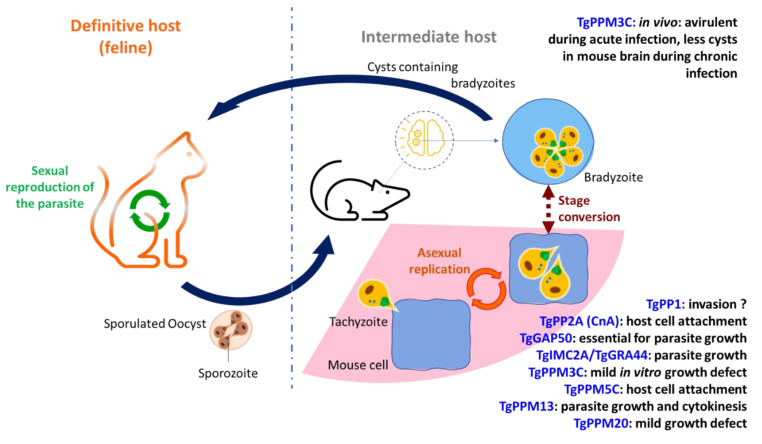
Schematic illustration of *T. gondii* life cycle showing the protein phosphatases with known stage-specific function (see Appendix A, Appendix A for details and references). This scheme has been created using PowerPoint from Microsoft Office 365.

**Table 1 microorganisms-10-00585-t001:** Comparison of the human (Hs) S/T/Y phosphatome composition to those of *Plasmodium falciparum* (*Pf*), *Babesia divergens* (*Bd*), *Toxoplasma gondii* (*Tg*) and *Cryptosporidium parvum* (*Cp*). Phosphatases are grouped in families following the classification from Chen et al. [24] (see Appendix A, Appendix A).

Fold/Superfamily	Family	Substrate	Number in Each Organism					
			Hs	*Pf*	*Pb*	*Bd*	*Tg*	*Cp*
PPPL (Phosphoprotein phosphatases-like)	PPP	pSer/pThr	13	14	13	6	15	10
	PAP	unknown	2	2	2	1	4	2
PPM (Protein phosphatase Mn^2+^ or Mg^2+^-dependent)	PPM	pSer/pThr	20	13	13	4	34	10
CC1 (Cysteine-based Class I)	PTP	pTyr	37	2	3	1	1	0
	DSP	pTyr, pSer/pThr	40	1	1	1	7	5
	DSP (RHOD)	pTyr, pSer/pThr	11	2	2	0	4	0
CC2 (Cysteine-based Class II)	LMWPTP	pTyr	1	1	1	0	2	2
	SSU72	pSer	1	0	0	0	0	0
CC3 (Cysteine-based Class III)	CDC25	pTyr, pThr	3	0	0	0	0	0
PTPL (Protein tyrosine phosphatase-like)	PTPLA *		0	1	1	1	1	1
HAD (Haloacid dehydrogenase)	EYA	pTyr	4	0	0	0	0	0
	FCP & NIF-like	pSer	8	4	5	6	9	5
Total			140	40	41	20	77	35

(*) *Plasmodium* PTPLA has been recently re-annotated as DEH (3-hydroxyacyl-CoA dehydratase) and may no longer be considered as a phosphatase (see below in the text).

## Data Availability

Not applicable.

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
