# Peer review of "Deciphering the Role of Protein Phosphatases in Apicomplexa: The Future of Innovative Therapeutics?"

_microorganisms, 2022, doi:10.3390/microorganisms10030585_

Round 1
Reviewer 1 Report
The present manuscript is a detailed review on protein phosphatases of apicomplexan parasites and their potential role as drug targets. The recent literature – in particular with respect to phosphoproteomic studies – is discussed in an appropriate manner.
The presentation of the text needs, however, to be improved with respect to a few points.
First of all the title is misleading in the sense that only protein phosphatases are discussed and not phosphatases in general. There are certainly some phosphatases involved e.g. in intermediate metabolism, which may be of interest. I therefore suggest to focus the title on protein phosphatases.
Concerning Fig1 and Table 1, it would be helpful to explain the abbreviations (PTPL, PPPL..) within the corresponding legends.
Some formulations are difficult to understand. A few examples:
Ll 85ff…while kinases fold follows a single structural model…phosphatases can be grouped..
Do the authors want to say…while kinases fold according to a single structural model…the folding of phosphatases differs according to their catalytic activities. Therefore, phosphatases can be grouped into different families..?
L.103..This reduced size…Do the authors want to say:.. The reduced number of phosphatases in Apicomplexa (or apicomplexan parasites) as compared to mammalian species…?
Ll 536 ff..An abundance of 118 phospho-peptides…Do the authors want to say: ..118 phospho-peptides were more abundant in the TgPPM3C knock-out strain than in the corresponding wildtype, 10 phospho-peptides were less abundant..
These are only a few examples. Overall, I recommend to submit the – otherwise excellent -manuscript to an native speaker before resubmission.
Author Response
Reviewer 1
The present manuscript is a detailed review on protein phosphatases of apicomplexan parasites and their potential role as drug targets. The recent literature – in particular with respect to phosphoproteomic studies – is discussed in an appropriate manner.
The presentation of the text needs, however, to be improved with respect to a few points.
First of all the title is misleading in the sense that only protein phosphatases are discussed and not phosphatases in general. There are certainly some phosphatases involved e.g. in intermediate metabolism, which may be of interest. I therefore suggest to focus the title on protein phosphatases.
We thank the reviewer for her/his positive feedback. We agree with his/her comment on the title and have modified it accordingly.
Concerning Fig1 and Table 1, it would be helpful to explain the abbreviations (PTPL, PPPL..) within the corresponding legends.
Explanations of the abbreviations of the families of phosphatases have been added in the legend of Figure 1 and completed in Table 1.
Some formulations are difficult to understand. A few examples:
Ll 85ff…while kinases fold follows a single structural model…phosphatases can be grouped..
Do the authors want to say…while kinases fold according to a single structural model…the folding of phosphatases differs according to their catalytic activities. Therefore, phosphatases can be grouped into different families..?
We have modified this sentence in the revised manuscript.
L.103..This reduced size…Do the authors want to say:.. The reduced number of phosphatases in Apicomplexa (or apicomplexan parasites) as compared to mammalian species…?
We have modified this sentence in the revised manuscript.
Ll 536 ff..An abundance of 118 phospho-peptides…Do the authors want to say: ..118 phospho-peptides were more abundant in the TgPPM3C knock-out strain than in the corresponding wildtype, 10 phospho-peptides were less abundant..
We have modified this sentence in the revised manuscript.
These are only a few examples. Overall, I recommend to submit the – otherwise excellent -manuscript to an native speaker before resubmission.
The manuscript has been carefully revised by a scientific native English speaker.
Reviewer 2 Report
Comment to the Authors
Obligate parasites contribute significant number of diseases all over the world every year. Specifically, Apicomplexa contributes substantial mortality and morbidity in the world. On top of that they add heavy economic burden on countries in the world. Drug resistance is on rise for anti-parasitic drugs. Therefore, development in drug discoveries in this area is an urgent need. In this important review paper authors Aline Freville, Benedicte Gnangnon, Asma S. Khelifa, Mathieu Gissot, Jamal Khalife and Christine Pierrot very nicely reviewed and discussed range of apicomplexan phosphatases and its potential therapeutic application to inhibit growth and virulence of these parasites. They have highlighted critical role of phosphatase in parasitic life cycle and its survival. Various phosphatomes comparison is provided with relevant references from related published literature. At the end authors also highlighted potential protein phosphatases drug targets.
Overall, in the field of public health, pharmacology, microbiology and biological sciences this review is important, and the research references cited by the authors are reasonable. However, the following points need to be addressed before publication, to make this article more helpful for readers of Microorganisms:
- Authors compared Serine/Threonine and Tyrosine for Plasmodium falciparum, Babesia divergens, Toxoplasma gondii and Cryptosporidium parvum with human phosphatome. It would be very informative for readers if authors also add comments on comparison of phosphatome for similar Apicomplexa which causes same disease. For example, Plasmodium vivax, malariae and ovale which also causes malaria like Plasmodium falciparum. What would be their phosphatome comparison with human phosphatome? What would be this comparison in case of their definitive hosts? For e.g., Mosquitoes is definitive host for Plasmodium required for their life cycle.
- It would be helpful for readers to understand if authors can explain or clarify the color code in Figure 1 label. Labeling color code for phosphates as done in the text will make it easier for readers to understand protein phosphatases differences between parasites and intermediate host. Similarly, it would be easier to read, if authors can define organisms’ names in footnote below, as it is defined in the text.
- Adding short discussion on advantages of drug targets on phosphatase as compared to other enzymes. Or adding a table in main text with possible phosphatase target sites, their characteristics and functions in the parasite would be very convenient to read the data discussed in the text. Are there any drugs currently being used to inhibit phosphatases targets, which are absent in humans and only present in parasites, discussed in this review? This will help increase significance of the new potential phosphatase sites. Are there any drug resistances against phosphatase inhibitors currently being used against parasites mentioned in this review? or if there is any potential for developing drug resistance and their probable causes? Such information would be very useful for the readers and further increase significance of this review.
- It would be also helpful if authors can summarize in one small paragraph on how this review will contribute further on to the development of current known anti-phosphatase agents? Such discussion would be really helpful and useful for the readers of Micororganisms and research community.
- There are already few review articles published recently about role of phosphatases in parasitic biological functions its virulence and its affect on host. For e.g., 2021 Gomez-Sandoval et. al. review in Parasitol. Res., 2008 Andreeva and Kutuzov review in Int. J. Parasitol. How this review is different from these reviews?
Minor corrections:
- Page 2 line 82, “(reviewed in [22]” the parenthesis is not closed.
- Page 9 line 355 space is missing between “ H ,allowing” and need to remove space between H and comma.

Author Response
Reviewer 2
Obligate parasites contribute significant number of diseases all over the world every year. Specifically, Apicomplexa contributes substantial mortality and morbidity in the world. On top of that they add heavy economic burden on countries in the world. Drug resistance is on rise for anti-parasitic drugs. Therefore, development in drug discoveries in this area is an urgent need. In this important review paper authors Aline Freville, Benedicte Gnangnon, Asma S. Khelifa, Mathieu Gissot, Jamal Khalife and Christine Pierrot very nicely reviewed and discussed range of apicomplexan phosphatases and its potential therapeutic application to inhibit growth and virulence of these parasites. They have highlighted critical role of phosphatase in parasitic life cycle and its survival. Various phosphatomes comparison is provided with relevant references from related published literature. At the end authors also highlighted potential protein phosphatases drug targets.
Overall, in the field of public health, pharmacology, microbiology and biological sciences this review is important, and the research references cited by the authors are reasonable. However, the following points need to be addressed before publication, to make this article more helpful for readers of Microorganisms:
- Authors compared Serine/Threonine and Tyrosine for Plasmodium falciparum, Babesia divergens, Toxoplasma gondii and Cryptosporidium parvum with human phosphatome. It would be very informative for readers if authors also add comments on comparison of phosphatome for similar Apicomplexa which causes same disease. For example, Plasmodium vivax, malariae and ovale which also causes malaria like Plasmodium falciparum. What would be their phosphatome comparison with human phosphatome? What would be this comparison in case of their definitive hosts? For e.g., Mosquitoes is definitive host for Plasmodium required for their life cycle.
We thank the reviewer for her/his positive feedback and comments on our review. In this review we have chosen to examine the protein phosphatome of reference Apicomplexa (including two Plasmodium species which were used in most studies) to highlight the specificities between them and compared to the mammalian host. Although interesting, deciphering the phosphatome of other species mentioned by the reviewer would be a massive undertaking since nothing has been published concerning phosphatases of these species and especially no data exist concerning their potential essentiality. We feel that including other phosphatases from other species is out of the scope of this review.
- It would be helpful for readers to understand if authors can explain or clarify the color code in Figure 1 label. Labeling color code for phosphates as done in the text will make it easier for readers to understand protein phosphatases differences between parasites and intermediate host. Similarly, it would be easier to read, if authors can define organisms’ names in footnote below, as it is defined in the text.
As recommended, we added the complete names of the protein phosphatases families in the figure legend.
- Adding short discussion on advantages of drug targets on phosphatase as compared to other enzymes. Or adding a table in main text with possible phosphatase target sites, their characteristics and functions in the parasite would be very convenient to read the data discussed in the text
To clarify this point, we did not mention that phosphatases would have advantages as drug targets compared to other enzymes. But we do believe that they will constitute additional and new drug targets. This has been clearly discussed in the introduction of chapter 4 of the submitted review. In addition, in chapter 5, we included a short discussion with perspectives. At this stage, we could not go further in details as nothing is known up to now about target sites of any phosphatases
Are there any drugs currently being used to inhibit phosphatases targets, which are absent in humans and only present in parasites, discussed in this review? This will help increase significance of the new potential phosphatase sites. Are there any drug resistances against phosphatase inhibitors currently being used against parasites mentioned in this review? or if there is any potential for developing drug resistance and their probable causes? Such information would be very useful for the readers and further increase significance of this review.
So far and to the best of our knowledge, there are no Phosphatases inhibitors which are used to treat any human disease or infection. As mentioned in this review, clinical trials are still undergoing and a new company was created recently to develop further such treatments and we assume that the resistance issues will be examined during development.
- It would be also helpful if authors can summarize in one small paragraph on how this review will contribute further on to the development of current known anti-phosphatase agents? Such discussion would be really helpful and useful for the readers of Micororganisms and research community.
This has been discussed in chapter 4 and 5. See also our answer to comment 3.
- There are already few review articles published recently about role of phosphatases in parasitic biological functions its virulence and its affect on host. For e.g., 2021 Gomez-Sandoval et. al. review in Parasitol. Res., 2008 Andreeva and Kutuzov review in Int. J. Parasitol. How this review is different from these reviews?
The review of Andreeva and Kutuzov (Int. J. Parasitol 2008) was focused only on tyrosine phosphatases and it is a bit dated. So no information on recent analysis of the function of the phosphatases are mentioned, mainly due to the recent introduction of reverse genetic analysis .
Regarding the review of Gomez Sandoval et al in 2021, it addresses the biological functions of some phosphatases with a focus on their potential role in virulence and evasion of host immune responses. In our review, we provide an overall view of all phosphatases in two major human apicomplexan parasites with the focus on how they could be used as drug targets. We also presented an updated overall classification of protein phosphatases in Apicomplexa.
Minor corrections:
- Page 2 line 82, “(reviewed in [22]” the parenthesis is not closed.
- Page 9 line 355 space is missing between “ H ,allowing” and need to remove space between H and comma.
These points have been corrected in the revised manuscript